# Plasma BRAF Mutation Detection for the Diagnostic and Monitoring Trajectory of Patients with LDH-High Stage IV Melanoma

**DOI:** 10.3390/cancers13153913

**Published:** 2021-08-03

**Authors:** Sofie H. Tolmeijer, Rutger H. T. Koornstra, Jan Willem B. de Groot, Maartje J. Geerlings, Dirk H. van Rens, Marye J. Boers-Sonderen, Jack A. Schalken, Winald R. Gerritsen, Marjolijn J. L. Ligtenberg, Niven Mehra

**Affiliations:** 1Department of Medical Oncology, Radboud Institute for Molecular Life Sciences, Radboud University Medical Center, 6525 GA Nijmegen, The Netherlands; sofie.tolmeijer@radboudumc.nl; 2Department of Medical Oncology, Radboud Institute for Health Sciences, Radboud University Medical Center, 6525 GA Nijmegen, The Netherlands; RKoornstra@Rijnstate.nl (R.H.T.K.); Dirk.vanRens@radboudumc.nl (D.H.v.R.); Marye.Boers-Sonderen@radboudumc.nl (M.J.B.-S.); Winald.Gerritsen@radboudumc.nl (W.R.G.); 3Department of Medical Oncology, Rijnstate Hospital, 6815 AD Arnhem, The Netherlands; 4Department of Medical Oncology, Isala Oncology Center, 8025 AB Zwolle, The Netherlands; j.w.b.de.groot@isala.nl; 5Department of Human Genetics, Radboud Institute for Molecular Life Sciences, Radboud University Medical Center, 6525 GA Nijmegen, The Netherlands; Maartje.Geerlings@radboudumc.nl (M.J.G.); Marjolijn.Ligtenberg@radboudumc.nl (M.J.L.L.); 6Department of Urology, Radboud Institute for Molecular Life Sciences, Radboud University Medical Center, 6525 GA Nijmegen, The Netherlands; Jack.Schalken@radboudumc.nl; 7Department of Pathology, Radboud Institute for Molecular Life Sciences, Radboud University Medical Center, 6525 GA Nijmegen, The Netherlands

**Keywords:** circulating tumor DNA (ctDNA), BRAF, biomarker, lactate dehydrogenase (LDH), S100, melanoma

## Abstract

**Simple Summary:**

For patients with metastatic melanoma, a rapid *BRAF* mutation assessment is vital to reveal the treatment options per patient. Additionally, close monitoring of the disease during treatment is essential to allow for adjustments in the treatment strategy when needed. The aim of this prospective study was to confirm the clinical validity of circulating tumor DNA (ctDNA) for minimally invasive *BRAF* mutation testing and treatment monitoring of metastatic melanoma patients with elevated lactose dehydrogenase serum levels. We observed that ctDNA-based *BRAF* mutation testing was a reliable and minimally-invasive alternative to tissue-based testing in 98% of all patients and was 100% specific. The changes in ctDNA levels during therapy appeared helpful for disease monitoring and outperformed other blood-based biomarkers in predicting treatment response.

**Abstract:**

For patients with newly diagnosed metastatic melanoma, rapid BRAF mutation (mBRAF) assessment is vital to promptly initiate systemic therapy. Additionally, blood-based biomarkers are desired to monitor and predict treatment response. Circulating tumor DNA (ctDNA) has shown great promise for minimally invasive mBRAF assessment and treatment monitoring, but validation studies are needed. This prospective study utilized longitudinal plasma samples at regular timepoints (0, 6, 12, 18 weeks) to address the clinical validity of ctDNA measurements in stage IV melanoma patients with elevated serum lactate dehydrogenase (LDH > 250U/L) starting first-line systemic treatment. Using droplet digital PCR, the plasma mBRAF abundance was assessed in 53 patients with a BRAFV600 tissue mutation. Plasma mBRAF was detected in 50/51 patients at baseline (98% sensitivity; median fraction abundance of 19.5%) and 0/17 controls (100% specificity). Patients in whom plasma mBRAF became undetectable during the first 12–18 weeks of treatment had a longer progression-free survival (30.2 vs. 4.0 months; *p* < 0.001) and cancer-specific survival (not reached vs. 10.2 months; *p* < 0.001) compared to patients with detectable mBRAF. The ctDNA dynamics outperformed LDH and S100 dynamics. These results confirm the clinical validity of ctDNA measurements as a minimally invasive biomarker for the diagnostic and monitoring trajectory of patients with LDH-high stage IV melanoma.

## 1. Introduction

Metastatic melanoma is the most aggressive and lethal form of skin cancer [1,2,3]. Two therapeutic approaches have become standard of care for this disease: targeted therapy and immunotherapy. Targeted therapy (BRAF/MEK inhibitor) has a rapid anti-tumor effect and is of benefit to the majority of patients. However, this therapy is limited to those harboring a *BRAFV600* mutation (mBRAF) in their tumor, and resistance commonly occurs within 12 months [4,5]. Immunotherapy, on the other hand, can achieve long-term disease control and is independent of *BRAF* status. Still, immunotherapy does not demonstrate sufficient anti-tumor activity in 50–70% of patients and is associated with a higher incidence of grade 3–4 toxicity [6,7,8,9]. 

To determine the best treatment strategy per patient, it is essential to rapidly assess mBRAF status and closely monitor treatment response. In current practice, mBRAF status is determined from routinely performed tumor biopsies, but this remains an invasive and time-consuming method. Treatment response is monitored by radiographic imaging, which limits frequent measurements and has difficulties distinguishing pseudo-progression from true progression, particularly following checkpoint inhibitors [10,11]. Therefore, alternative strategies have been investigated to improve current practice, including the use of circulating tumor DNA (ctDNA). ctDNA is released into the bloodstream by apoptosis and necrosis of tumor cells [12], enabling the detection of mBRAF from blood. In addition, easily obtainable repetitive blood draws allow close monitoring of ctDNA dynamics in relation to treatment response. 

Previous studies established ctDNA analysis as a highly specific tool for mBRAF detection, but ctDNA-based mBRAF detection can vary in sensitivity (56–90%) [13,14,15,16,17,18,19]. The varying sensitivity can be explained by the ctDNA quantity in patients, which is in turn dependent on the tumor burden and location of the tumor. For instance, in patients with M1c disease, it appeared 2–5 times more likely to detect ctDNA compared to M1a/b disease [20,21,22]. In addition, elevated lactate dehydrogenase (LDH), associated with tumor cells outgrowing their blood supply, is associated with 30–50 times higher ctDNA levels [14,23,24]. Consequently, the sensitivity of ctDNA-based mBRAF assessment can vary per patient, hampering the implementation of ctDNA-based mBRAF assessment in routine patient care. 

Besides the diagnostic application of ctDNA for mBRAF assessment, the ctDNA burden is prognostic for patient outcome. Similar to LDH, the amount of ctDNA at the start of treatment appears prognostic for the progression-free survival (PFS) and overall survival (OS) of patients with metastatic melanoma [16,17,21,25,26]. Additionally, changes in ctDNA were shown to be relevant for the monitoring of treatment response. For example, a conversion of ctDNA from detectable to undetectable levels during immunotherapy or targeted therapy was shown to reflect a 3–7 times longer PFS and 4-8 times longer OS [16,18,21,27]. Small and retrospective studies indicate that ctDNA outperforms the other blood-based biomarkers for melanoma, LDH and S100, in predicting patient outcome [15,28,29].

To validate the current applications for ctDNA, prospective clinical validation studies are needed using blood samples at regular time points and standardized blinded assessment of outcome parameters. Syeda and colleagues published the first large clinical validation study showing the potential of ctDNA as an independent biomarker for targeted therapy in patients with advanced melanoma [26]. Plasma mBRAF was detected in 93% (320/345) of all patients using droplet digital PCR (ddPCR) [26]. In patients with elevated LDH levels, the sensitivity was 98%, showing a great promise for ctDNA-based mBRAF assessment, particularly in LDH-high stage IV melanoma patients. As elevated LDH is associated with a 50% shorter OS compared to patients with normal LDH [30], prompt initiation of treatment and close treatment monitoring is essential for these patients. Interestingly, Syeda and colleagues observed a better predictive value of ctDNA dynamics for the PFS and OS in LDH-high patients compared to LDH-normal patients [26]. Unfortunately, longitudinal sampling beyond 4 weeks was missing in this study, and the ctDNA dynamics were not compared to other blood-based biomarkers. 

The current study aimed to confirm and expand on the clinical validity of ctDNA measurements for diagnostic and monitoring trajectory of patients with LDH-high metastatic melanoma starting their first-line of systemic treatment. Utilizing longitudinal and prospectively collected plasma samples at fixed timepoints up to 18 weeks of treatment, accompanied by radiographic imaging and evaluation of other blood-based biomarkers, we aim to elaborate on the potential of ctDNA measurements for systemic treatment monitoring in LDH-high stage IV melanoma patients.

## 2. Materials and Methods

### 2.1. Patient Cohort and Study Design

Patients with stage IV metastatic melanoma were enrolled in the study between March 2017 and June 2020. All patients had a confirmed BRAFV600 mutation (mBRAF) in tissue based on routine diagnostic tests. All patients had elevated serum LDH (>250 U/L) and were naïve for both immune checkpoint blockade agents and BRAF/MEK inhibitors. Written consent was obtained from all patients as approved by the local medical ethical committee (dossier number 2016–2769, December 2016). Patients underwent baseline characterization, including physical examination, blood marker evaluation, and radiographic tumor assessment. Patients started with either BRAF/MEK inhibitors or immune checkpoint blockade upon inclusion and underwent clinical evaluation every 6 weeks, which included blood collection and CT scans. CT results were assessed by RECIST 1.1 criteria [31], which distinguishes between complete response (CR), partial response (PR), stable disease (SD), or progressive disease (PD). 

In order to confirm the specificity of ctDNA-based mBRAF detection, blood was also collected from three LDH-high melanoma patients without mBRAF in their tumor and 14 healthy controls. 

### 2.2. Cell-Free DNA Isolation and ctDNA Quantification

Blood was collected at baseline and after 6, 12, and 18 weeks of treatment using EDTA tubes. Within 4 hours, the blood samples were first centrifuged at 120× *g* for 20 min to separate plasma from blood cells. Afterward, plasma was centrifuged at 360× *g* for 20 min to remove platelets. Finally, the plasma was centrifuged at 14,000× *g* for 10 min to remove cellular debris. Plasma was stored at −80°C until further processing. 

Total cell-free circulating DNA was extracted from approximately 2 mL of plasma using the QIAamp Circulating Nucleic Acid Kit (Qiagen) according to the manufacturer’s protocol and eluted in 30 µL low-TE buffer. The DNA concentration was quantified using Qubit (ThermoFisher), and the quality was checked on a Fragment Analyzer (Agilent high sensitivity genomic DNA kit #DNF-488-0500). Next, the presence of mBRAF ctDNA copies was assessed with the droplet digital PCR (ddPCR) BRAFV600 screening kit (#12001037, BioRad), which can detect BRAF p.V600E (c.1799T>A), p.V600R (c.1798_1799delinsAG), and p.V600K (c.1798_1799delinsAA) mutations. All samples were measured in duplicate. A binominal distribution was used to calculate the theoretical sensitivity of detecting mBRAF per sample based on the available input material (Appendix A). Samples with two or more mutant droplets were considered ctDNA positive. 

To convert cell-free DNA concentration-units from ng per mL plasma to copies per mL plasma, we multiplied the concentrations by a factor of 303, assuming that the mass of a haploid genome is 3.3 pg. Subsequently, the ctDNA copies per mL plasma could be calculated based on the fractional abundance of mBRAF and the total cell-free DNA copies per mL plasma.

### 2.3. Statistical Analysis

The correlation between continuous variables was calculated using Spearman rank correlation statistics. Differences in ctDNA levels concerning the absence or presence of specific metastasis sites were calculated using an unpaired two samples Wilcoxon test. Time-to-event outcomes, including PFS and melanoma cancer-specific survival (CSS), were described via the Kaplan–Meier method. PFS and CSS were defined as the time from the start of therapy to the date of first reported PD for PFS and death as a consequence of melanoma for CSS. PFS and CSS curves were stratified according to patient characteristics and clinicopathologic features and compared using Cox-regression models. For the Cox-regression models, the baseline ctDNA copies were log-transformed for a normal distribution and assessed as a continuous variable. For assessment of ctDNA dynamics in longitudinal samples, ctDNA results were dichotomized as detectable (positive) or undetectable (negative) after 12–18 weeks of treatment. S100 and LDH dynamics were also dichotomized as below the upper limit of normal or above the upper limit of normal after 12–18 weeks. Due to missing data and a limited number of events for CSS (*n* = 16), a multivariable Cox-regression was only used to evaluate PFS. This multivariate Cox-regression analysis included all variables significantly associated with PFS in the univariate analysis (*p* < 0.05).

## 3. Results

### 3.1. Patient Characteristics

A total of 53 patients with LDH-high metastatic melanoma were included in this study (Table 1). Half of these patients were treated with combination immunotherapy (ipilimumab + nivolumab), while the other half was first treated with combination BRAF/MEK inhibitors before starting with immunotherapy. As this translational work is part of an ongoing clinical trial, we cannot disclose patient treatment specifics. The median follow-up duration was 12.3 months (range 0–38.1 months). Fifty-eight percent of patients were alive at the time of analysis, and 42% had an ongoing treatment response.

### 3.2. ctDNA-Based mBRAF Assessment

In total, 153 blood samples were collected for ctDNA analysis (data available in Appendix A). This included a baseline sample for 51/53 (96%) patients and longitudinal follow-up for 40/53 (75%) patients. Of all baseline plasma samples, mBRAF was detected in 50/51 (98%) with a median fractional abundance of 19.5% (range 0.2–66.5%). The one patient for whom mBRAF could not be detected had M1b disease and the smallest cumulative RECIST target lesions of the cohort. 

Figure 1A visualizes the baseline plasma mBRAF abundance in relation to other baseline characteristics, such as LDH levels and metastasis sites. mBRAF abundance moderately correlated with levels of LDH (Figure 1B, ρ = 0.50, *p* < 0.001), weakly correlated with S100 levels (Figure 1C, ρ = 0.35, *p* = 0.03), and strongly correlated with total cell-free circulating DNA (Figure 1D, ρ = 0.83, *p* < 0.001). No association was found between the mBRAF levels and the RECIST sum of the target lesion diameters (SLD) (*p* = 0.74), but higher mBRAF levels were observed in patients with liver metastasis (Figure 1E, *p* = 0.05).

To determine the specificity of ctDNA-based mBRAF detection, plasma of 17 controls was tested for the presence of mBRAF. Fourteen of these controls were healthy individuals, and three were patients with LDH-high metastatic melanoma but without mBRAF in their tumor. All the plasma samples tested negative for mBRAF (Appendix A). Combined, this indicates that ctDNA-based mBRAF detection has a specificity of 100% and sensitivity of 98% in LDH-high stage IV melanoma patients.

### 3.3. ctDNA Dynamics and Treatment Response

After treatment initiation, we investigated ctDNA dynamics in relation to the treatment response. For 40 patients, follow-up plasma samples were available up to 12–18 weeks and/or at progression. An overview of the longitudinal blood-biomarker assessments available per patient is given in Appendix A. The ctDNA dynamics of these patients could be divided into two groups: (1) 23 patients in whom plasma mBRAF became undetectable during the first 12–18 weeks of treatment, including the patient who was tested mBRAF-negative at baseline, and (2) 17 patients in whom plasma mBRAF remained detectable (or became detectable again) during the first 12–18 weeks of treatment. Figure 2 visualizes the ctDNA dynamics in both groups, referred to as (1) ctDNA negative and (2) ctDNA positive. Appendix A includes the results on the S100 and LDH dynamics in the ctDNA dynamics groups.

In the ctDNA negative group, only three (13%) patients experienced disease progression within 18 weeks. All three patients had an ongoing response per RECIST1.1 of their target lesions but developed one or more new lesions. One of the three patients developed only one new lesion that was located in the brain. Four (17%) patients in the ctDNA negative group had disease progression after 18 weeks, and 16 (70%) had an ongoing treatment response at the time of analysis. In the ctDNA positive group, 14 (82%) patients developed disease progression within 18 weeks and 2 (12%) after 18 weeks. Only one (6%) patient had a continuing treatment response. The ctDNA content of this patient was still declining from baseline to the last measured timepoint.

### 3.4. ctDNA Dynamics Associates with the PFS and CSS

Next, we investigated the ctDNA dynamics in relation to PFS and CSS. Figure 3A demonstrates that patients with undetectable ctDNA after 12–18 weeks of treatment had a 7.4 times longer median PFS compared to patients with still detectable ctDNA (30.2 vs. 4.0 months; hazard ratio (HR) 12.6 (95% confidence interval [95% CI] 4.3–36.8)). A similar difference was observed for CSS (Figure 3B; not reached vs. 10.2 months; HR 14.6 (95% CI 3.3–64.6)). 

Other parameters that were significantly associated with a shorter PFS in the univariable analysis included the presence of liver metastasis, the amount of mBRAF copies at baseline, and the S100 dynamics (being above or under the upper limit of normal after 12–18 weeks) (Table 2). Interestingly, only ctDNA dynamics remained significant in a multivariable model (Table 2). The univariable analysis for CSS revealed that similar variables associated with a shorter PFS were also associated with a shorter CSS (Appendix A). Due to limited events in the CSS analysis and missing data (Appendix A), we did not perform a multivariable analysis with all significantly associated variables for CSS. Still, ctDNA dynamics was the strongest prognostic variable in the univariate analysis for both PFS and CSS.

## 4. Discussion

The current study confirmed that ctDNA could be a valuable diagnostic and predictive tool in patients with LDH-high metastatic melanoma. Before treatment initiation, ctDNA-based mBRAF assessment was shown to be highly sensitive and specific for these patients. Using prospectively collected longitudinal data at fixed timepoints, it was shown that ctDNA dynamics can be used to monitor treatment response. The ctDNA dynamics defined favorable and unfavorable profiles that could be used as an independent predictor of long-term response and survival and may eventually be used to guide treatment adaptations.

To determine optimal therapeutic choices in metastatic melanoma, knowledge on the *BRAF* status is vital. Particularly in newly diagnosed, symptomatic, LDH-high metastatic melanoma, prompt treatment initiation with BRAF/MEK inhibition allows for a rapid tumor and clinical response in patients harboring mBRAF. At present, *BRAF* status is assessed using a tumor biopsy, but ctDNA-based mBRAF detection could become a new standard being a less-invasive and faster strategy for accurate *BRAF* assessment (within days instead of weeks). With the level of ctDNA in blood to a large extent dependent on tumor burden [4,20,21], we hypothesized that ctDNA-based mBRAF detection would be most reliable in patients with LDH-high metastatic disease. Elevated LDH is associated with tumors outgrowing their blood supply [32] and has clinical utility in melanoma. LDH is used as a classifier for the American joint committee on cancer staging [33] and is a strong prognostic biomarker for metastatic melanoma independent of treatment [34,35]. Moreover, previous studies revealed a clear correlation between LDH and ctDNA levels (R = 0.50–0.76) [14,15,16,23]. 

A moderate correlation between LDH and ctDNA was still observed in this study even though patients were preselected for elevated LDH levels (ρ = 0.50). More importantly, 50/51 patients positive for mBRAF by tumor tissue test also had detectable mBRAF in plasma resulting in a 98% sensitivity of the ctDNA-based assay. This is similar to the results of Syeda and colleagues, who also observed a 98% sensitivity among 125 patients with LDH-high advanced melanoma [26]. The sensitivity is higher compared to previous studies in which patients were not treatment-naïve or preselected for elevated LDH (56–90%) [13,14,15,16,17,18,19]. The sensitivity was independent of the assay threshold that was previously reported to affect sensitivity [18]. The specificity obtained in this study (100%) was comparable to other studies [13,14,15,16,17,18,19,36]. As approximately 40% of all metastatic melanoma patients have elevated serum LDH [4], ctDNA-based mBRAF detection can become a reliable alternative to tissue-based testing for a substantial number of patients to guide the initial choice of systemic therapy. The next step for the implementation of ctDNA-based mBRAF testing in the clinic would be to offer ctDNA-based mBRAF testing in parallel to tissue testing and compare the turn-around time, sensitivity, and costs. 

Besides the baseline ctDNA detection, the ctDNA changes in relation to treatment response were investigated in this study. Figure 2 illustrates that the majority of patients in whom ctDNA became undetectable during the first 12-18 weeks (ctDNA negative) had a favorable treatment response. Disease progression within 18 weeks was only observed in three patients. Interestingly, all these patients had responding RECIST target lesions but developed one or more new lesions. In patient 20, the new lesion was located in the brain, potentially explaining the absence of ctDNA in plasma [18,21,37]. Patient 26 was later diagnosed with myelofibrosis, which might explain the new lesions on the CT-scan without mBRAF detection in blood. This patient was switched to BRAF/MEK inhibitors after the new lesions were observed and had a complete response following the next 2.5 years, which is a remarkable duration of response to BRAF/MEK inhibitors. In contrast to the ctDNA negative group, most patients in the ctDNA positive group experienced disease progression within the first 18 weeks of treatment. Only one patient had a long-term treatment response beyond 18 weeks. This patient showed an ongoing decline in ctDNA copies from baseline in all measured timepoints and a longer ctDNA evaluation period may therefore have resulted in undetectable mBRAF level following the 18-week period. Altogether, the data suggest that longitudinal measurements of ctDNA during treatment could help monitor treatment response. 

When translating the observations from Figure 2 to a survival analysis, a clear association was observed between ctDNA dynamics and time to progression or death. Patients with undetectable levels of ctDNA after 12–18 weeks of treatment had a 7.6 time longer median PFS compared to patients with detectable ctDNA (Figure 3A). Moreover, only two melanoma-related deaths were observed in this group following a median follow-up of 18.4 months and included the death of patient 20, who developed brain metastasis (Figure 3B). These observations are in line with previous literature, describing comparable hazard ratios for the ctDNA detectability at 3 to 12 weeks [18,21,27]. Similarly, around 50% of the patients with detectable ctDNA at the start of treatment convert to undetectable ctDNA after a few weeks of treatment [18,27]. 

Importantly, ctDNA dynamics improved discrimination between progressing and non-progressing patients within the observation period of 18 weeks when compared to S100 and LDH dynamics (Appendix A; Appendix A; Table 2). This is in line with results from other small studies and retrospective studies [15,28,29]. Overall, the ctDNA dynamics resembled S100 dynamics despite a weak correlation between the variables at baseline. Nonetheless, S100 dynamics misclassified five patients that were correctly classified by ctDNA dynamics (Appendix A). LDH dynamics misclassified nine patients and was not significantly associated with PFS in the univariate analysis (Appendix A; Appendix A). Only ctDNA dynamics remained significant in a multivariate Cox-regression model for PFS after correction for the significant baseline characteristics and S100 dynamics (Table 2). Therefore, ctDNA dynamics seems a very specific and valuable biomarker to distinguish long-term responders from patients who may benefit from an early switch to an alternative treatment strategy or possibly treatment intensification. Ongoing randomized trials are evaluating whether ctDNA-informed early treatment switch may improve outcome compared to standard of care; this includes a randomized phase II trial in patients with metastatic melanoma treated with dabrafenib and trametinib, where following a predefined ctDNA drop of ≥80%, patients are switched to nivolumab and ipilimumab (NCT03808441). In the adjuvant setting, a phase 3 randomized, blinded trial will investigate whether ctDNA-informed early treatment initiation with nivolumab has a superior outcome to the standard of care in patients following resection of stage IIB/C melanoma (NCT04901988).

To conclude, this study on prospectively collected material underlines the potential of ctDNA assessment as a diagnostic and predictive tool for patients with LDH-high metastatic melanoma. Limitations of this study included the small number of patients and a ctDNA follow-up that stopped at 18 weeks. Future studies will be needed to investigate the clinical utility of ctDNA-based mBRAF assessment in routine practice and help identify optimal use of longitudinal ctDNA follow-up. Nevertheless, this study supports the next steps in the implementation of ctDNA assessments in routine clinical care of metastatic melanoma patients. 

## 5. Conclusions

In conclusion, the current study confirmed the clinical validity of ctDNA-based mBRAF detection as an alternative to tissue-based testing for patients with LDH-high metastatic melanoma. Using prospectively collected blood samples at regular timepoint, the study underlines the potential of ctDNA dynamics to monitor and independently predict treatment response. 

## Figures and Tables

**Figure 1 cancers-13-03913-f001:**
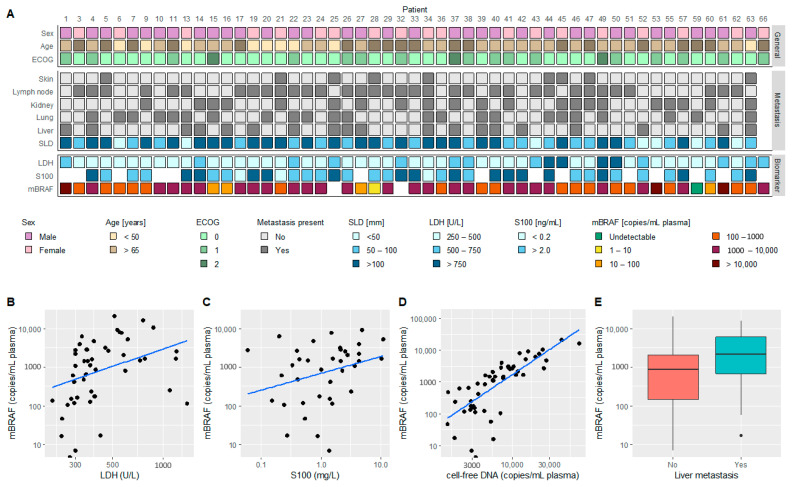
Baseline patient characteristics and clinicopathologic features in relation to baseline plasma mBRAF copies. (**A**) Schematic overview illustrating the patient characteristics (sex, age, ECOG), metastasis sites and blood-based biomarkers (LDH, S100) relative to the plasma mBRAF copies; (**B**) Correlation between plasma mBRAF copies and LDH levels (ρ = 0.50, *p* < 0.001); (**C**) Correlation between plasma mBRAF copies and S100 levels (ρ = 0.35, *p* = 0.03); (**D**) Correlation between plasma mBRAF copies and total cell free circulating DNA copies (ρ = 0.83, *p* < 0.001); (**E**) Association between plasma mBRAF copies and the presence of liver metastasis (*p* = 0.05).

**Figure 2 cancers-13-03913-f002:**
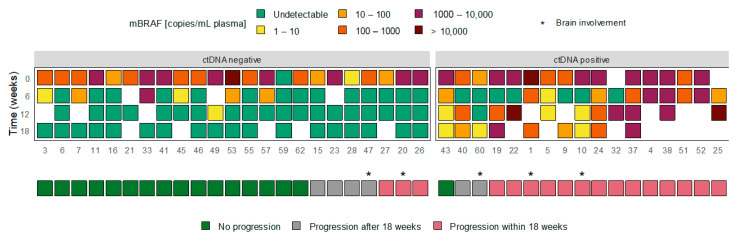
Overview of the ctDNA dynamics (mBRAF copies/mL plasma) related to treatment response within 18 weeks. Patients were stratified according to their ctDNA levels at 12–18 weeks, being either detectable (ctDNA positive) or undetectable (ctDNA negative).

**Figure 3 cancers-13-03913-f003:**
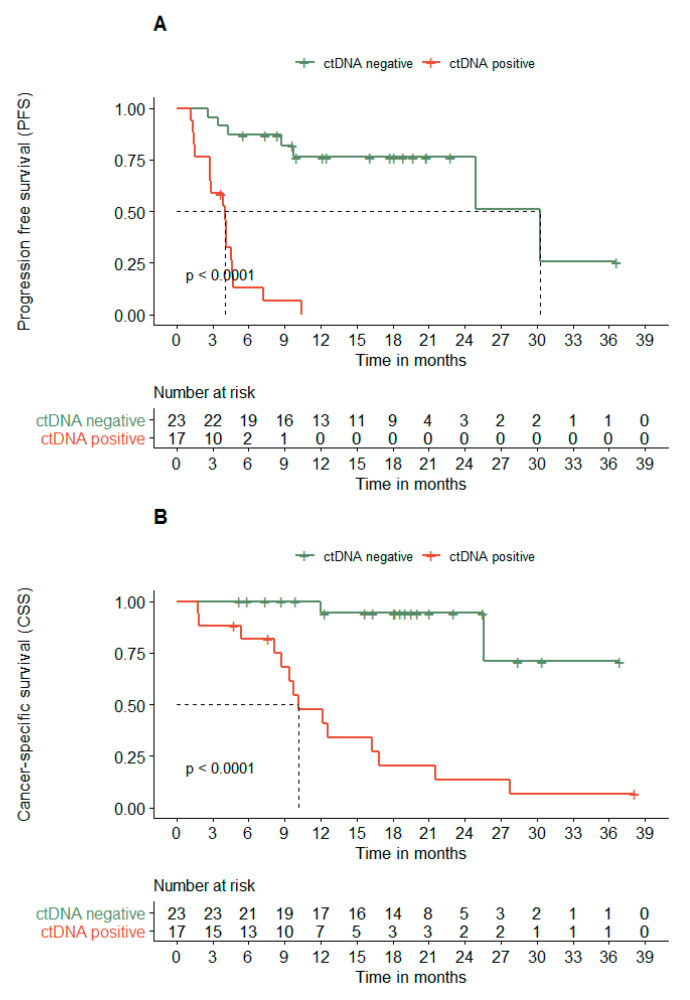
ctDNA dynamics related to (**A**) progression-free survival (PFS) and (**B**) cancer-specific survival (CSS). Patients were stratified according to their ctDNA levels at 12–18 weeks, being either detectable (ctDNA positive) or undetectable (ctDNA negative).

**Table 1 cancers-13-03913-t001:** Baseline patient characteristics.

Total Patients, *n* (%)	53 (100%)
Sex, *n* (%)	
Female	19 (36%)
Male	34 (64%)
Age	
Median years (range)	61 (28–78)
ECOG, *n* (%)	
0	34 (64%)
1	16 (30%)
2	3 (6%)
Initiated treatment, *n* (%)	
Immunotherapy	28 (53%)
BRAF/MEK inhibitor	25 (47%)
LDH (U/L)	
Median (range)	357 (261–1560)
S100 (ng/mL)	
Median (range)	1.43 (0.06–10.97)
Metastasis location, *n* (%)	
Skin	9 (17%)
Lymph node	34 (64%)
Lung	23 (43%)
Kidney	21 (40%)
Liver	19 (36%)
Follow-up	
Median months (range)	12.3 (0–38.1)

**Table 2 cancers-13-03913-t002:** Factors associated with the progression-free survival (PFS). HR = hazard ratio, 95% CI = 95% confidence interval, ULN = upper limit of normal.

		Progression-Free Survival					
		Univariate Analysis			Multivariate Analysis		
Variable		HR	95% CI	*p*-Value	HR	95% CI	*p*-Value
Liver metastasis	Present vs. absent	3.65	1.73–7.71	<0.001	1.38	0.36–5.3	0.643
ctDNA baseline	log10 (mBRAF per mL plasma)	1.68	1.07–2.64	0.02	1.26	0.67–2.4	0.473
S100 dynamics	Above vs. below ULN at 12-18 weeks	5.5	2.21–13.71	<0.001	0.95	0.23–4.0	0.94
ctDNA dynamics	Positive vs. negative at 12-18 weeks	12.57	4.30–36.76	<0.001	18.75	3.55–98.9	<0.001

## Data Availability

The data presented in this study are available in Appendix A.

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
