# Peer review of "Plasma BRAF Mutation Detection for the Diagnostic and Monitoring Trajectory of Patients with LDH-High Stage IV Melanoma"

_cancers, 2021, doi:10.3390/cancers13153913_

Round 1

Reviewer 1 Report

This is a well-prepared article reflecting a high level of work done. The results are presented in a good quality. The results of the study may be of interest for the prompt choice of therapies (targeted therapy or immunotherapy) for patients with melanoma with high levels of lactate-dehydrogenase as well as for monitoring response to therapy based on cfDNA dynamics and other biomarkers.

The topic of the article is not new. In the literature there are quite a lot of studies related to the detection of BRAF mutations in plasma cfDNA in various forms of cancer, including melanoma:

https://pubmed.ncbi.nlm.nih.gov/32026754/

https://www.ncbi.nlm.nih.gov/pmc/articles/PMC4837559/

https://www.thelancet.com/journals/lanonc/article/PIIS1470-2045(20)30726-9/fulltext

https://pubmed.ncbi.nlm.nih.gov/32330187/

https://www.ncbi.nlm.nih.gov/pmc/articles/PMC6679157/

https://mct.aacrjournals.org/content/15/6/1397

https://www.annalsofoncology.org/article/S0923-7534(19)44769-8/fulltext

In contrast to a number of similar reports, this work was done with a smaller group of patients.

However, such studies are needed to confirm the possibility of using new biomarkers in the diagnosis of cancer. Digital PCR is a sensitive technology for detecting rare copies of circulating DNA. The technology is comparable in simplicity to real-time PCR and allows non-invasive testing.

Given the scale of the problem of finding new biomarkers, the general design of the study, and the detailed rationale for the status of biomarkers in melanoma patients (LD, ctDNA, BRAFV600), I recommend this article for publication in its current form.

Author Response

We thank reviewer for the constructive comments regarding our manuscript. We acknowledge that plasma BRAFm detection in metastatic melanoma and other tumour types is not new, but are thankful of your consideration that our prospective clinical study in 53 patient builds on previous publications and is of additional value to the field. 

Reviewer 2 Report

This manuscript provides important confirmation and extension of previous work looking at the value of testing blood samples for ctDNA for mutated Braf in patients with advanced melanoma.  In particular, they obtain longitudinal samples from advanced melanoma patients whose LDH is elevated, a key marker of high tumor burden.  In this group, they confirm earlier reports that both sensitivity and specificity are exceptionally high.  They carefully correlate ctDNA dynamics at 12-18 weeks with PFS and cancer specific survival, showing a hazard ratio of 12.6 (for PFS), which provides a much stronger association than clinical markers, including LDH level itself.

This is potentially very important to clinical practice.  Targeted therapy works rapidly in almost all patients with BRAF mutated advanced melanoma, but the duration of response is limited, especially in those with high tumor burden. Because of the rapid reliable response, even in patients who may be too ill to tolerate or respond to immune therapy, many such patients are started on targeted therapy.  A real time assay of response and impending relapse would greatly aid clinicians in the common and critical decision of how to time the switch from targeted therapy to immune therapy.   The data presented here support the idea that ctDNA may be such an assay.

My only specific suggestion is to include discussion of the need to switch from targeted therapy to immune therapy and possibly also reference ongoing Phase III studies on sequencing of therapy.

Author Response

We thank reviewer for carefully reviewing our manuscript and for the constructive comments. We also are grateful for your confirmation that our prospective clinical study in 53 patient, builds on previous literature, and is of additional value to the field. We have followed your suggestion, and included in the discussion, a paragraph regarding a phase 2 (NCT03808441) and 3 trial (NCT04901988) in melanoma, that are utilizing a ctDNA-informed treatment switch, and comparing this to standard care (or blinded to ctDNA results).